# An Examination of Barriers to Accessing Mental Health Care, and Their Association with Depression, Stress, Suicidal Ideation, and Wellness in a Bangladeshi University Student Sample

**DOI:** 10.3390/ijerph20020904

**Published:** 2023-01-04

**Authors:** Munjireen Sifat, Maisha Huq, Mirza Baig, Naima Tasnim, Kerry M. Green

**Affiliations:** 1TSET Health Promotion Research Center, University of Oklahoma Health Science Center, Oklahoma City 73105, OK, USA; 2Department of Behavioral & Community Health, University of Maryland, College Park, MD 20742, USA; 3American Psychiatric Group, Baltimore, MD 21202, USA; 4Institute of Governance and Development, BRAC University, Dhaka 1212, Bangladesh

**Keywords:** suicide, depression, young adults, barriers to mental healthcare, stigma

## Abstract

Background: The mental health burden is high and rising among Bangladeshi university students. Understanding barriers to mental healthcare and how barriers impact mental health outcomes may inform the development of targeted interventions to decrease barriers and improve access to care. Aims: This study identifies barriers to mental healthcare and their association with mental health outcomes in a Bangladeshi university student sample. Methods: We conducted a cross-sectional survey (*n* = 350) on stigma-related, attitudinal, and instrumental barriers to accessing mental healthcare among Bangladeshi university students. We examined the association between stigma and non-stigma (i.e., attitudinal and instrumental) barriers with four mental health outcomes: suicidal ideation, depression, high perceived stress, and wellness. Results: Attitudinal barriers were the most reported barriers. Stigma-related barriers were significant for individuals who had experienced suicidal ideation (aOR = 2.97, *p* = 0.001), not for individuals with depression. Non-stigma-related barriers were significant for individuals who had experienced depression (aOR = 2.80, *p* = 0.011). Conclusions: The current work advances our understanding of how to improve access to mental healthcare among university students in Bangladesh. Stigma-related barriers were particularly salient for individuals who experienced suicidal ideation. Further study is needed on how stigma may impact access to care distinctly for different mental health problems among Bangladeshi university students.

## 1. Introduction

The prevalence of mental disorders has increased worldwide across demographics, cultures, and political contexts [1]. According to the World Health Organization, the COVID-19 pandemic triggered a 25% rise in depression and anxiety worldwide (2 March 2022). Low and middle-income countries face a particularly high mental health burden, in part due to their lack mental healthcare resources. One such low-income country is Bangladesh, which has one of the lowest rates of psychiatrists in the world with one psychiatrist per 200,000 people [2,3] and a suicide rate of over three times that of the world (10.7/100,000) [4].

It is likely that mental health is worsening among Bangladeshi university students, especially since the onset of the COVID-19 pandemic. A study of 544 university students in Bangladesh found that about 78% of the students were afflicted with mental disorders during the COVID-19 lockdown [5]. Another cross-sectional investigation of 403 undergraduate students of the International University of Business Agriculture and Technology in Bangladesh [6] found that 33.7% of the sample met the criteria for depression. Despite the reliance on convenience samples, these findings indicate that Bangladeshi university students are at high risk for depression. 

However, there has been little research on the barriers to mental healthcare and their relationship to mental health problems among Bangladeshi university students. Stigma is cited as one of the most prevalent barriers to accessing mental healthcare, including negative help-seeking attitudes (e.g., embarrassment) and personally held prejudices about mental illness and treatment [7]. In many low-income countries, addressing mental health is challenging due to existing beliefs that mental health conditions result from disobedience of religion, culture, or family [8], and the associated stigma [9]. Professional mental health services are less likely to be utilized when these beliefs are held [10]. It is crucial to assess and understand attitudes and beliefs about mental health in low-income countries in order to increase the utilization of mental health services.

The current study asks, “What is the prevalence of attitudinal, instrumental, and stigma-related barriers to mental healthcare in university students of Bangladesh?” We also examine the association between these barriers to mental healthcare and suicidal ideation, depression, perceived stress, and mental wellness. Understanding the prevalence of barriers to care and the association between barriers to accessing mental healthcare and mental health outcomes may inform the development of targeted interventions to decrease barriers, thus allowing more people to seek care.

## 2. Materials & Methods

### 2.1. Survey Development

This study was granted ethical approval by the University of Maryland Institutional Review Board (#1656046-3) and all participants provided consent for their data to be used in the research. Data were stored on University of Maryland’s secure, password-encrypted, network. 

A survey in English was created, and consultation with native Bangla speakers was done for a culturally competent Bangla translation of the survey. Cognitive interviews (*n* = 5) were conducted to allow participants in the target demographic to explain their interpretations of survey questions and define mental health. A pilot test (*n* = 10) was conducted to identify items that were difficult to understand or culturally inappropriate.

### 2.2. Study Sample

To be eligible for the study, participants had to meet the following criteria: (1) be 18 years or older, (2) be currently enrolled in a university located in Bangladesh. Faculty from five universities were contacted to assist with participant recruitment, and four university faculty agreed to actively partake in recruitment efforts, reflecting the majority of responses from Jahangirnagar University (62.8% of participants), Bangladesh University of Business and Technology (6%), East West University (4.4%), and Dhaka University (4.1%). Faculty of the universities emailed invitations to students to complete the anonymous online survey. Several invitations were posted on social media platforms, including Facebook pages for current university students. Over the course of two months, 370 valid responses were recorded to the survey. Each participant had the option to be entered into a raffle, with a one-in-three chance of receiving 5 USD.

### 2.3. Data Handling

The instrument included a validity question to ensure that participants read questions accurately. One question was added toward the last third of the survey to assess if students were reading the questions closely when answering survey items, the question read, “Please choose option C”, only data from participants who selected C were used when analyzing data.

### 2.4. Statistical Analysis

Descriptive statistics were used to report participants’ demographic characteristics, including their age, gender, sexual orientation, childhood socioeconomic background, relationship status, year of schooling, degree of study, and university attended.

A priori alpha level was set at 0.05. R^2^ was reported for the adjusted logistic regression model to explain the total variance. Analyses were conducted using the statistical package SPSS (SPSS 25, 2017). Separate regression models examined the relationship between stigma, attitudinal and instrumental barriers to care, and mental health outcomes. These examined the association between the distinct barriers to care and mental health outcomes. Regression models examining the relationship between stigma and non-stigma (i.e., attitudinal and instrumental) barriers to care, and mental health outcomes were also assessed. 

Pearson’s correlations and collinearity diagnostics were used to assess possible multicollinearity between independent variables for each model; this found that the constructs within the BACE scale were highly correlated (r = 0.52–0.68, *p* < 0.01). The correlation between stigma and non-stigma (i.e., attitudinal and instrumental) barriers was also high (r = 0.76), which highlighted the need to separate constructs in analysis in order to avoid multicollinearity. Accordingly, we ran separate models examining (i) each barrier individually (including stigma, attitudinal, and instrumental barriers as individual barriers) in relation to the outcomes, and (ii) including both stigma and non-stigma constructs in the same model.

Adjusted odds ratios were reported for the model, along with model fit statistics. Reliability analysis was also included, using Cronbach’s alpha to report the reliability of scales used in the study. Two-tailed significance was reported for all analyses. 

### 2.5. Measures

#### 2.5.1. Demographic Characteristics

Gender was categorized as one identifying as (1) male, (2) female, and (3) gender minority. Age was measured as a continuous variable. Sexual orientation was categorized as (1) straight/heterosexual or (2) sexual orientation minority, which included identifying as gay or lesbian, bisexual, asexual, uncertain or questioning, or preferring to self-describe with the option to write their answer. Relationship status was measured as a categorical variable: (1) single, (2) in a relationship or married, (3) other, which consisted of divorced, separated, or widowed, prefer to self-describe. Family socioeconomic status was measured by asking, “While growing up, how often did your family have enough money to make ends meet?” with responses measured on a Likert scale of 0 = never and 5 = always. Time enrolled in university was measured by asking what year of study they were in (1–4+). One item, how religious do you consider yourself, measured self-perception of religiosity on a 1–10 scale, with higher scores meaning greater religiosity.

#### 2.5.2. Dependent Variables

***Subjective psychological well-being.*** was measured using a 5-item HERO Wellness Scale. This wellness scale has high reliability (Cronbach’s alpha of 0.87 in this sample) and assesses positive psychological traits such as optimism, resilience, happiness, and enthusiasm. Questions such as, “On average, during the last seven days, how happy have you felt?” are measured using a 10-point Likert scale from 0 = not at all happy to 10 extremely happy. Adding the values of five answers on the scale was used to calculate the total raw score. The total possible score ranged from 0 to 50. This score was then dichotomized into low and moderate levels of wellness (scores 0 through 33) and high levels of wellness (scores 34 and above) based on a 75th percentile cut-off to assess high scores relative to the sample scores. 

***Perceived Stress.*** A 4-item self-report instrument, the Perceived Stress Scale (PSS-4), developed by Cohen et al., (1983) [11], was used to assess stress among students. This scale measures stress using Likert scale responses (0 = never 4 = very often) to questions such as “In the last month, how often have you felt that you were unable to control the important things in your life?” Cronbach’s alpha of this scale in this sample was acceptable at 0.70. Scale scores summed to 0–16, and where dichotomized, with a score of 0–9 considered a low to moderate level of stress and scores of 11–16 considered a high level of stress. 

***Depression.*** The Patient Health Questionnaire-2 (PHQ-2) was used to measure depressive symptoms [12]. Participants answer the questions on a scale from 0 = never to 3 = almost every day. Depression severity is measured by summating the responses (range 0–6) to questions. A score of 3 is the optimal cut point indicating that major depressive disorder is likely. Depressive symptoms were categorized as high (3 or more points) and low scores (0–2 points).

***Suicidal Ideation.*** Suicidal ideation was measured using one yes/no item, “Have you ever had thoughts that you would rather be dead or having thoughts about hurting yourself in some way”?

#### 2.5.3. Independent Variables

***Barriers to Accessing Mental Healthcare.*** The Barriers to Access to Care Evaluation scale (BACE) scale has 30 items that examine both stigma and non-stigma related to mental health service utilization subscales [13], encompassing anticipated social stigma, stigma by association, internalized stigma, disclosure concerns, stereotypes, and discrimination. Participants responded using a Likert-scale response 0 = not at all (indicating this was not a barrier to care), 3 = a lot (indicating a great barrier to care), to the following question, “Have any of these issues ever stopped, delayed or discouraged you from getting, or continuing with, professional care for a mental health problem?” See Table 1.

The 12-item stigma subscale of the Barriers to Access to Care Evaluation scale (BACE) was used to assess perceived stigma [13] (Cronbach’s α = 0.89), an example item being “Concern about what people at work might think, say or do”. A mean was calculated for the 12 items.

In the 12-item-subscale for instrumental barriers, participants rated statements about instrumental barriers to getting mental health services on a Likert scale, (0) not a barrier to (3) a lot (major barrier). Instrumental barrier statements included “Not being able to afford the financial costs involved” and “Being unsure where to get professional help”. Four additional items were added to the scale to include virtual therapy, for example, “Lack of private space for virtual therapy”, and issues specific to college students, for example, “Too many academic demands”. The final instrumental barrier subscale had a Cronbach’s alpha of 0.81 and was a mean of the 12 items. 

Finally, 10 items assessed peoples’ attitudinal barriers toward accessing care, for example, “Wanting to solve the problem on my own” and “Thinking the problem would get better by itself”. The reliability of this construct was alpha = 0.71 and the overall score was a mean of the 10 items.

## 3. Results

### 3.1. Descriptive Results

Table 2 reports participants’ demographic characteristics. The mean age was 22.7 years old (SD = 1.9), 59.2% were male, and the majority (93.9%) reported being heterosexual. The mean score on the Hero Wellness Scale was 26.12 (range = 0–50, SD = 10.37), Perceived Stress Scale mean was 8.46 (range = 0–16, SD = 3.4), the sample had a mean depressive score of 2.6 (range = 0–6, SD = 1.69). Twenty-eight percent of the sample reported having experienced suicidal ideation. Of stigma (Mean = 0.64, SD = 0.63), attitudinal (mean = 1.10, SD = 0.50), and instrumental (mean = 0.62, SD = 0.55) barriers to mental healthcare, attitudinal barriers were the most common.

Attitudinal barriers were reported to be the most common barriers by our respondents. “Wanting to solve the problem on my own” was the single largest barrier among all the barriers according to our sample, with 60% reporting it stopped, delayed, or discouraged them from getting, or continuing with, professional care for a mental health problem a lot of the time. On average, our respondents endorsed stigma barriers slightly more than instrumental barriers. 

#### 3.1.1. Instrumental Barriers

Looking at instrumental barriers, 36.6% of respondents reported not being able to afford the financial cost of care as a barrier ‘quite a lot’ or ‘a lot;’ 33.2% of respondents reported that not knowing where to find professional help is ‘quite a lot’ or ‘a lot’ of a barrier. 

#### 3.1.2. Attitudinal Barriers

“Wanting to solve the problem on my own” was the most endorsed barrier, with 87.9% of the respondents saying this was ‘quite a lot’ or ‘a lot’ of an attitudinal barrier to access mental healthcare; 76.5% of respondents reported “thinking the problem would get better by itself” as ‘quite a lot’ or ‘a lot’ of a barrier, which was the second most common attitudinal barrier. “Preferring to get help from family or friends” was reported to be the third most common barrier by 50.8% of respondents as ‘quite a lot’ or ‘a lot’ of a barrier. 

#### 3.1.3. Stigma Barriers

Examining the stigma barriers, 45.5% of respondents reported “concern about what my family might think, say, do or feel” as ‘quite a lot’ or ‘a lot’ of a barrier; this was the most endorsed barrier among stigma barriers. “Feeling embarrassed or ashamed” was the second most common stigma barrier, reported by 35.5% of the respondents as ‘quite a lot’ or ‘a lot’ of a barrier. “Concern that I might be seen as weak for having a mental health problem” was reported to be the third most common stigma barrier by 34.6% of the respondents as ‘quite a lot’ or ‘a lot’ of a barrier. 

### 3.2. Logistic Regression Results

#### 3.2.1. Depression

Table 3 reports the logistic regression model results of the relationships between stigma, attitudinal and instrumental barriers to care, and depression. In the models of each individual barrier’s relationship with depression, all three barriers were significantly and positively associated with greater depression after adjusting for confounders. A one-point increase in stigma barriers was associated with a 2.4 times greater likelihood of depression (*p* < 0.001), and a one-point increase in non-stigma barriers was associated with a four times greater likelihood of depression (*p* < 0.001), holding constant select demographic characteristics. In the fully adjusted model, including both stigma and non-stigma-related barriers, stigma barriers were no longer significantly associated with depression (*p* = *0*.217). In contrast, a one-point increase in non-stigma barriers was associated with a 2.8 times greater likelihood of depression (*p* = 0.011). The fully adjusted model, including both stigma and non-stigma-related barriers, explained 14.7% (Nagelkerke R^2^) of the variance in depression.

#### 3.2.2. Perceived Stress

The logistic regression results of the associations between barriers to mental healthcare and high perceived stress are also shown in Table 3. Although each barrier was positively associated with perceived stress in the individual models, they were not significant in the fully adjusted model. For example, although a one-point increase in score of stigma barriers was associated with a 2.17 times greater likelihood of high perceived stress (*p* < 0.001) in the individual model, stigma barriers were not significant (*p* = 0.199) in the fully adjusted model. Similarly, while a one-point increase in non-stigma barriers was associated with a 3.17 times greater likelihood of high perceived stress (*p* < 0.001) in the individual model, non-stigma barriers were not significant in the fully adjusted model (*p* = 0.087). The fully adjusted model, including both stigma and non-stigma-related barriers, explained 10.8% (Nagelkerke R^2^) of the variance in perceived stress.

#### 3.2.3. Suicidal Ideation

Table 4 reports the associations between stigma, attitudinal and instrumental barriers to mental healthcare, and suicidal ideation. In the individual models of the barriers’ relationship with suicidal ideation, all three barriers were significantly positively associated with greater suicidal ideation. A one-point increase in endorsing stigma barriers was associated with a 2.93 times greater likelihood of suicidal ideation (*p* < 0.001); and a one-point increase in non-stigma barriers was associated with a 3.03 times greater likelihood of suicidal ideation (*p* < 0.001) holding constant select demographic characteristics. In the fully adjusted model, including both stigma and non-stigma-related barriers, only stigma barriers were significantly associated with suicidal ideation (*p* = 0.002). The fully adjusted model, including both stigma and non-stigma-related barriers, explained 23.5% (Nagelkerke R^2^) of the variance in suicidal ideation. 

#### 3.2.4. Wellness

In the individual models of the barriers’ relationships with wellness, stigma (aOR = 0.43, *p* = 0.001) and non-stigma (aOR = 0.41, *p* = 0.006) barriers were statistically significantly, inversely associated with greater wellness (Table 5). Only stigma barriers remained significant in the fully adjusted model, including both stigma and non-stigma-related barriers (*p* = 0.043). The model, including both stigma and non-stigma-related barriers, explained less than 0.1% (Nagelkerke R^2^) of the variance in wellness.

## 4. Discussion

The current work aimed to (i) describe barriers to accessing mental healthcare among Bangladesh University students, and (ii) explore the associations between these barriers and four mental health outcomes, depression, suicidal ideation, perceived stress and wellness. The level of poor mental health in this sample is high and consistent with prior studies of depression and suicide among this group during the pandemic [14,15], notwithstanding the reports of varied prevalence rates of mental health conditions among Bangladeshi students in previous work. The findings regarding the association between barriers and mental health outcomes were particularly intriguing.

The five most commonly reported barriers to mental healthcare in this sample were: wanting to solve the problem by oneself; thinking the problem would get better by itself; preferring to get help from family or friends; being concerned about what family might think, say, or do; and not being able to afford the cost of care. The most commonly reported barriers in this study generally mirrored those reported in prior studies—albeit limited—based in Bangladesh [16] as well as those of young adults from other cultural contexts [17,18,19,20], which find attitudinal and stigma to be the most common barriers. 

Compared to stigma-related and instrumental barriers, attitudinal barriers were reported most frequently. This can be surprising as past research notes high levels of stigma toward mental illness in Bangladesh [21] and given the country’s mental healthcare system supply is vastly under-resourced [22]. Still, instead of stigma and instrumental barriers, the most endorsed barrier in our sample remained attitudinal-“wanting to solve the problem on my own”.

Self-reliance has also been found to be a top barrier among university students and young adults in the USA [17,19] and UK [18], and adults in China [20]. Thus, self-reliance may be a common reason that individuals do not access mental healthcare across cultural contexts. Past studies suggest self-reliance may also be a common barrier across various age groups [23]. It is possible that students who endorse self-reliance may not perceive that their mental health needs are severe enough to need outside help. Indeed, previous work in this sample found that the perception of needing help for mental health predicted the use of clinical services [24]. Current findings show that raising awareness about the benefits of professional care over self-reliance may improve access to mental healthcare in this population, which is supported by research showing that people who have more positive views of clinical mental health are more likely to use such services [24].

Among attitudinal and stigma-related barriers, family-level factors may be more salient among Bangladesh university students than students in other cultural contexts. Concern about what family might think was the most endorsed stigma-related barrier in this sample. In contrast, concern about what family may think is a less frequent barrier among US university students, with 11% reporting this as a major barrier in a US sample [13] compared to the 45.5% of our sample of Bangladeshi students who reported it as a quite a lot or a lot of a barrier.

Therefore, although several anti-stigma programs from high-income settings have been implemented in low- and middle-income countries [24,25,26,27,28] there may be a need to adapt existing anti-stigma programs to address family-level factors in Bangladesh. Fostering social environments that encourage the recognition of needing help is important as it may not only influence their attitudes toward seeking care (reduce attitudinal barriers) but also may reduce the likelihood of experiencing depression itself. While preferring family as a source of support was a top attitudinal barrier in this sample and prior research among university students, future interventions should guide Bangladeshi families on how to facilitate, instead of impede, professional care for students.

Further research is needed to understand the role of stigma-related barriers to care for individuals experiencing suicidal ideation versus depression. Intriguingly, we found stigma-related barriers significantly predicted suicidal ideation but not depression in the fully adjusted models in this sample. The findings may comport three lines of prior research: First, individuals who have attempted suicide experience more internal stigma relative to individuals who have not attempted suicide [29,30]. Second, public stigma is greater toward individuals with suicidal ideation than individuals with depressed mood [31]. Further research is needed to assess whether internal and/or public stigma against suicide-related care seeking is in fact greater than stigma against seeking care for depressed individuals in Bangladesh.

Third, advances in anti-stigma discussions may have reduced the role of stigma on care-seeking specifically for depression in the past two decades. A study found there has been a reduction in depression stigma during the 2006–2018 period in the USA after holding constant covariates according to a time, age, and cohort analysis [32]. Increased information and story-sharing through social media may have also reduced depression stigma [33]. Further research is needed to assess if depression-related stigma may have reduced in Bangladesh, bringing attitudinal and instrumental barriers to the fore. 

Despite the lack of mental health professionals in Bangladesh, the most endorsed instrumental barrier for this sample was the cost of services in lieu of the availability of services. The high cost of mental health services may be why students of this sample were interested in using digital health for mental health support [34,35], as digital health provides a no to low-cost service for mental health promotion. 

### 4.1. Implications

Future research should explore how mental healthcare access barriers –measured at a more detailed level—vary by mental health outcome. While the current work explored barrier categories (i.e., stigma-related, non-stigma-related) by mental health outcome, understanding which barrier items (e.g., concern about family) are most salient for each mental health outcome may inform how to design future interventions. Future interventions should also address self-reliance and family-level factors to improve access to mental healthcare among Bangladesh university students. Finally, research should examine coping strategies that protect participants from poor mental health.

### 4.2. Limitations

The current work has certain limitations. The BACE items to measure barriers to accessing mental healthcare were not initially developed for a Bangladeshi sample. It is possible that cultural or religious items could have been more relevant barriers. Secondly, we did not include protective factors for accessing mental healthcare. Understanding which factors facilitated mental healthcare access could have provided a more holistic understanding of improving access to mental healthcare. Thirdly, as the PHQ-2 comprises two items only and is used for screening purposes, it does not operationalize depression symptomology as holistically as larger scales such as the Center for Epidemiologic Studies Depression Scale [36]. As such future studies should examine depression using detailed scales for a more in-depth understanding of this outcome.

## 5. Conclusions

The current work advances our understanding of how to improve access to mental healthcare in Bangladesh. Despite the mental healthcare system in Bangladesh being vastly under-resourced, the primary barriers to accessing mental healthcare were stigma-related and attitudinal among university students. Family-level attitudinal and stigma-related factors were also found to be relevant to improving this population’s access to care. The findings also call for further study on how stigma may impact access to care distinctly for different mental health conditions among Bangladeshi university students.

## Figures and Tables

**Table 1 ijerph-20-00904-t001:** Types of Barriers to Accessing Care (% of sample endorsing barrier).

BACE Item	Not at All a Barrier	A Little Bit of a Barrier	Quite a Lot of a Barrier	A Lot of a Barrier
	%	%	%	%
**Instrumental Barriers**				
Being unsure where to go to get professional help	51.8	15.0	25.1	8.1
Problems with transport or traveling to appointments	74.5	12.6	8.0	4.9
Not being able to afford the financial costs involved	43.8	19.5	19.5	17.1
Professionals from ethnic/cultural group not available	83.3	6.5	6.5	3.6
Being too unwell to ask for help	82.6	9.8	5.5	2.1
Difficulty taking time off work	67.6	16.6	10.1	5.7
Having problems with childcare while I receive professional care	84.0	8.2	4.1	3.7
Having no one who could help me get professional care	67.4	14.2	13.3	5.1
Having too many academic demands	63.6	16.2	12.0	8.1
Too many non-academic/personal time commitments	52.2	19.4	20.1	8.3
Lack of private space for virtual therapy	58.5	17.3	17.9	6.4
Lack of technological resources to facilitate participation	70.3	13.6	10.2	5.9
**Attitudinal Barriers**				
Wanting to solve the problem on my own	5.5	6.6	27.6	60.3
Fear of being put in hospital against my will	84.3	7.6	5.5	2.6
Thinking the problem would get better by itself	23.5	17.1	31.5	27.9
Preferring alternative (traditional/religious healing) care	42.9	18.7	26.3	12.1
Thinking that professional care would not help	64.9	17.3	12.6	5.3
Dislike of talking about my feelings, emotions or thoughts	34.6	19.9	26.1	19.4
Concerns about the effects of the treatments available	48.2	24.2	19.4	8.2
Having had previous bad experiences	85.1	5.5	6.8	2.6
Preferring to get help from family or friends	30.1	19.1	37.1	13.7
Thinking I did not have a problem	43.8	17.8	24.1	14.4
**Stigma-related Barriers**				
Concern that I might be seen as weak for having a mental health problem	50.4	15.0	21.6	13.0
Concern that it might harm my chances when applying for jobs	75.8	10.1	8.4	5.7
Concern about what my family might think, say, do or feel	36.2	18.3	24.3	21.2
Feeling embarrassed or ashamed	43.1	21.4	23.7	11.8
Concern that I might be seen as ‘crazy’	77.1	10.3	8.8	3.8
Concern that people I know might find out	63.2	17.1	14.4	5.4
Concern that people might not take me seriously if they found out I was having professional care	70.0	14.1	10.5	5.4
Not wanting a mental health problem to be on my medical records	59.1	14.5	16.0	10.4
Concern about what my friends might think, say or do	61.1	19.0	16.0	3.9
Concern about what people at work might think, say or do	69.4	14.1	12.8	3.6

**Table 2 ijerph-20-00904-t002:** Participant Characteristics (*n* = 350).

	**%/M (SD)**
Age (18–41) ^a^	22.7 (1.86)
Gender	
Male	59.2%
Female and Gender Minority	40.8%
Sexual Orientation (% Heterosexual/Straight)	93.9%
Childhood SES (% Low SES)	28.3%
Relationship Status (% Non-partnered)	76.8%
Year/Semester in School ^b^	
1st–3rd/First year	20.6%
4th–6th/Second year	19.4%
7th–9th/Third year	20.0%
10th–12th/Fourth year	19.4%
12th+/Fourth year+	20.6%
Religiosity (1–10)	6.33 (2.00)
Wellness (0–50)	26.12 (10.37)
Wellness (% High)	24.9%
Perceived Stress (0–16)	8.46 (3.41)
Perceived Stress (% High)	28.0%
Depressive Symptoms (0–6)	2.60 (1.69)
High Depressive Symptoms (>3)	43.4%
Suicidal Ideation (Lifetime) ^c^	28.0%
Barriers to Care: Stigma (0–3)	0.64 (0.63)
Barriers to Care: Non-Stigma (0–3)	0.86 (0.45)
Barriers to Care: Attitudinal (0–3)	1.10 (0.50)
Barriers to Care: Instrumental (0–3)	0.62 (0.55)

Note: Superscripts denote a lower sample size due to missing data ^a^: *n* = 345, ^b^: *n* = 348, ^c^: *n* = 315; Religiosity ranges from 0 (not religious) to 10 (most religious); Barriers to care range from 0 (not at all) to 3 (a lot) of a barrier to seeking clinical care; Barriers to care: non-stigma was created by taking the mean of attitudinal and instrumental barriers to care.

**Table 3 ijerph-20-00904-t003:** Logistic Regression Associating Barriers to Accessing Mental Healthcare with Outcomes of Perceived Stress (*n* = 323) and Depression (*n* = 323).

	Individual Models of Each Barriers’ Relationship with Depression		Model Including Both Stigma and Non-Stigma-Related Barriers	
**Outcome: Perceived Stress (High vs. Low), *n* = 323**				
	OR (95% CI)	*p*	aOR (95% CI)	*p*
Stigma Barriers to Care	2.17 (1.46, 3.21)	<0.001	1.47 (0.82, 2.65)	0.199
Non-Stigma Barriers to Care (Mean of Instrumental and Attitudinal Barriers)	3.17 (1.79, 5.62)	<0.001	2.10 (0.90, 4.91)	0.087
Instrumental barriers to care	2.41 (1.52, 3.82)	<0.001	-	
Attitudinal barriers to care	2.35 (1.39, 3.98)	0.001		
**Nagelkerke R squared**	−		0.108	
*p*	−		0.003	
**Outcome: Depression (Yes vs. No), *n* = 323**				
Stigma Barriers to Care	2.41 (1.63, 3.58)	<0.001	1.42 (0.81, 2.49)	0.217
Non-Stigma Barriers to Care (Mean of Instrumental and Attitudinal Barriers)	4.01 (2.28, 7.05)	<0.001	2.80 (1.26, 6.22)	0.011
Instrumental barriers to care	2.58 (1.64, 4.05)	<0.001		
Attitudinal barriers to care	3.08 (1.86, 5.10)	<0.001		
**Nagelkerke R squared**	−		0.147	
*p*	−		<0.001	

Note: All models control for age, gender, sexual orientation, relationship status, year in school, socioeconomic status, and religiosity levels;; Attitudinal and instrumental barriers to care constructs were combined into “non-stigma barriers to care” due to collinearity (r = 0.52, *p* < 0.001; Statistically significant controls for perceived stress outcome: none; Statistically significant controls depression outcome: relationship status (partner vs. non-partnered) AOR = 0.48 (95% Confidence Interval = 0.27, 0.84), *p* = 0.010, and religiosity AOR = 0.902 (95% Confidence Interval = 0.80, 1.02) *p* = 0.010.

**Table 4 ijerph-20-00904-t004:** Logistic Regression Associating Barriers to Accessing Mental Healthcare with Suicidal Ideation, *n* = 293.

	Individual Models of Each Barriers’ Relationship with Depression		Model Including Both Stigma and Non-Stigma-Related Barriers	
	OR (95% CI)	*p*	aOR (95% CI)	*p*
Stigma Barriers to Care	2.93 (1.85, 4.62)	<0.001	2.97 (1.49, 5.92)	0.002
Non-Stigma Barriers to Care (Mean of Instrumental and Attitudinal Barriers)	3.03 (1.63, 5.63)	<0.001	0.97 (0.38, 2.51)	0.952
Instrumental barriers to care	2.29 (1.37, 3.81)	0.001	-	
Attitudinal barriers to care	2.41 (1.36, 4.30)	0.003		
**Nagelkerke R squared**	−		0.235	
*p*	−		<0.001	

Note: All models additionally control for age, gender, sexual orientation, relationship status, year in school, socioeconomic status, and religiosity levels; Attitudinal and instrumental barriers to care constructs were combined into “non-stigma barriers to care” due to collinearity; Statistically significant controls: religiosity (0–10) AOR = 0.79 (0.68, 0.91), *p* = 0.001 and gender (male reference) AOR = 2.83 (1.64, 4.86), *p* < 0.001.

**Table 5 ijerph-20-00904-t005:** Logistic Regression Associating Barriers to Accessing Mental Health Care with Wellness, *n* = 323.

	Individual Models of Each Barriers’ Relationship with Depression		Model Including Both Stigma and Non-Stigma-Related Barriers	
	OR (95% CI)	*p*	aOR (95% CI)	*p*
Stigma Barriers to Care	0.43 (0.26, 0.72)	0.001	0.48 (0.24, 0.98)	0.043
Non-Stigma Barriers to Care (Mean of Instrumental and Attitudinal Barriers)	0.41 (0.22, 0.77)	0.006	0.80 (0.32, 1.97)	0.629
Instrumental barriers to care	0.68 (0.41, 1.15)	0.150	-	
Attitudinal barriers to care	0.38 (0.21, 0.66)	0.001		
**Nagelkerke R squared**	−		<0.001	
*p*	−		0.140	

Note: All models additionally control for age, gender, sexual orientation, relationship status, year in school, socioeconomic status, and religiosity levels; Attitudinal and instrumental barriers to care constructs were combined into “non-stigma barriers to care” due to collinearity; Statistically significant controls: religiosity (0–10) AOR = 1.22 (95% CI = 1.06, 1.41), *p* = 0.007 and gender (male reference) AOR = 0.51 (95% CI = 0.29, 0.88), *p* = 0.016.

## Data Availability

De-identified data may be available on reasonable request, in conjunction with an executed data use agreement. All coding generated during the current study are not publicly available but may be requested from the corresponding author.

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
