# Peer review of "An Examination of Barriers to Accessing Mental Health Care, and Their Association with Depression, Stress, Suicidal Ideation, and Wellness in a Bangladeshi University Student Sample"

_ijerph, 2023, doi:10.3390/ijerph20020904_

Round 1

Reviewer 1 Report

The manuscript presents a study examining barriers to accessing mental health care and their association with four psychological outcomes in Bangladeshi University student sample.

Figure 1 is not clear enough for me. Additionally, some frequencies number are repeated in the text, unnecessarily.

The methods assessing outcomes, especially depression are unfortunately too superficial to make firm conclusions based on the results.

Author Response

We thank you for taking out time to review our submission. Please see the attachment. 

Reviewer 2 Report

Thank you for this interesting study, which is well described and presented.

Please add one sentence to the data security strategy which has been used in this study.

Additionally, for better reading/unterstanding I would recommend to transfer Figure 1 into a table due to the high amount of items.

Author Response

(The authors gave the same response as above.)

Reviewer 3 Report

Well constructed and piloted survey with consideration of subject understanding and interpretation of survey questions. Consideration of different recruitment strategies, and control of confounding variables through statistical modelling. 

Minor items to address:

-Needs clarification of how many Universities were contacted and how many agreed to post the survey out to student bodies.

-Need clarification of inclusion/exclusion criteria

-Need clarification of ‘validity question’ how did researchers’ structure and interpret this?

-Figure 1 missing title and formatted so that it sits below the figure

Author Response

(The authors gave the same response as above.)
